# Shenqi Fuzheng injection reverses M2 macrophage-mediated cisplatin resistance through the PI3K pathway in breast cancer

Bin Yan[1☯], Rong Shi[2☯], Yi-yu Lu[1], Dong-dong Fang[1], Mei-na Ye[3], Qian-mei Zhou[1]*

1 Institute of Interdisciplinary Integrative Medicine Research, Shanghai University of Traditional Chinese Medicine, Shanghai, China, 2 Shuguang Hospital Affiliated to Shanghai University of Traditional Chinese Medicine, Shanghai, China, 3 Longhua Hospital Affiliated to Shanghai University of Traditional Chinese Medicine, Shanghai, China

☯ These authors contributed equally to this work.
* tazhou@163.com

**Data Availability Statement:** All relevant data are within the paper.

**Funding:** This work was financially supported by a grant from the National Natural Science Foundation

## Abstract

### Background

Shenqi Fuzheng injection (SQFZ) combined with chemotherapy can sensitize tumour cells. However, the mechanisms underlying SQFZ's effects remain unknown. In human breast cancer cell lines and M2 macrophages, we showed that SQFZ was a significantly potent agent of sensitization.

### Methods

The human breast cancer cell line, MDA-MB-231/DDP, and the human acute leukaemia mononuclear cell line, THP-1, were used. MDA-MB-231/DDP breast cancer xenografts were established to monitor tumour growth. Resistance-associated proteins were examined by western blotting. Levels of cytokines and chemokines were detected by ELISA. Cell viability was measured using the MTT assay. Apoptosis was detected by flow cytometric analysis.

### Results

SQFZ significantly enhanced the capability of cisplatin to reduce tumour mass. SQFZ and cisplatin decreased the expression of CD206 by 1.89-fold and increased that of CD86 by 1.76-fold as compared to cisplatin alone. The levels of PGE2, IL-6, and CCL1 decreased significantly, and the activation of p-PI3K and the expressions of P-gp and ABCG2 were also inhibited by SQFZ in combination with cisplatin treatment in vivo. The survival following cisplatin administration of 60 µM and 120 µM reduced significantly in the presence of SQFZ in MDA-MB-231/DDP and M2 co-cultured cells. IGF-1, a PI3K activator, combined with SQFZ weakened the effects of SQFZ-induced apoptosis from 28.7% to 10.5%. The effects of IGF-1 on increasing the expressions of P-gp, ABCG2, and Bcl-2, and decreasing that of Bax were reversed by SQFZ.

of China (81803934). The funders had no role in study design, data collection and analysis, decision to publish, or preparation of the manuscript.

**Competing interests:** The authors have declared that no competing interests exist.

## Conclusion

Our findings provide evidence that SQFZ is a potential therapeutic drug for cancer therapy.

## Introduction

One out of ten women is diagnosed with breast cancer annually, and breast cancer is the second leading cause of death among women worldwide [1]. Various treatments have been used in clinical settings, including surgery, radiotherapy, and chemotherapy, along with hormones and targeted- and immunotherapy. However, drug resistance can lead to therapeutic failure and recurrence in these patients [2]. Drug resistance is a limiting factor for the cure of cancer patients [3], and numerous mechanisms underlying drug resistance have been revealed. Cisplatin (DDP) is a major chemotherapeutic agent used for the treatment of triple-negative breast cancer (TNBC). However, a risk of developing resistance to cisplatin treatment exists, and leads to treatment failure [4].

Abnormal expression of apoptosis-related factors leads to the destruction of apoptosis, resulting in cell death tolerance and drug resistance. The most thorough mechanism is transporter-mediated drug efflux. ATP-binding cassette (ABC) transporters utilize ATP to efflux several compounds across cellular membranes. Specifically, P-glycoprotein (P-gp/ABCB1), multidrug-resistant protein-1 (ABCC1/MRP1), and breast cancer resistance protein (ABCG2/BCRP) are significantly expressed in patients with TNBC [5]. Antitumour drugs are transported across the cell membrane, and it is, therefore, necessary to decrease the activity of efflux pumps, including ATP-ABC transporters.

The PI3K/Akt pathway enhances the biological basis of cancer by effectively expressing ABC transporters, including P-gp (ABCB1), MRP1 (ABCC1), and BCRP (ABCG2). Their activation may decrease the chemotherapeutic responses and increase drug efflux [6, 7]. The PI3K/Akt pathway is an important signalling cascade for MDR in breast cancer [8, 9], and survival signals can protect cancer cells from death. To reverse drug resistance induced by transport proteins, some mechanisms implicating the PI3K signalling pathways have been assessed.

Macrophages in the tumour microenvironment are commonly known as tumour-associated macrophages (TAMs) [10]. These can transform either into M1 and M2 types [11], whereby the former exhibit antitumour properties and express abundant CD86 [12], while the latter are associated with tumour progression [13] and express high levels of CD206 [14]. TAMs promote cancer cell proliferation, invasion, and metastasis in breast cancer [15]. Moreover, TAM-mediated resistance can be reversed by androgen blockade therapy in patients with prostate cancer [16]. TAMs activate the PI3K/Akt/mTOR signalling pathway to promote resistance in breast cancer [17]; however, its mechanistic details remain largely unclear.

In China, to enhance chemosensitivity and reduce chemotherapeutic resistance, traditional Chinese medicinal herbs have been widely used in breast cancer [18, 19]. Among them, the Shenqi Fuzheng injection (SQFZ) is developed from an extract of *Radix Astragali* and *Radix Codonopsis* mixed in a ratio of 1:1 to obtain an injectable preparation [20]. Consequently, the combination of SQFZ and either cisplatin or other chemotherapeutic drugs is used for breast cancer [21], lung cancer [22], and other malignant tumours [23].

Clinical trials have demonstrated the efficacy of SQFZ combined with chemotherapy in sensitizing tumour cells and lowering toxicity [24]. SQFZ can also reduce the side effects of chemotherapy and improve the quality of life of patients with cancer [25]. However, the underlying mechanisms of SQFZ-mediated reversal of chemoresistance and increased

chemotherapeutic sensitivity remain unknown. In this study, we aimed to elucidate the effects and molecular mechanisms underlying the action of the Chinese herbal formula, SQFZ, in vitro and in vivo.

## Materials and methods

### Cell culture and reagents

The human breast cancer cell line, MDA-MB-231/DDP, was donated by Longhua Hospital Affiliated with the Shanghai University of Traditional Chinese Medicine. The human acute leukaemia mononuclear cell line, THP-1, was obtained from the Shanghai Cell Biological Institute of the Chinese Academy of Science (Shanghai, China). These cell lines were revived from the stock in a medium without DDP and then cultured in a medium supplemented with DDP (5 μmol/L) the next day at 37°C and 5% $CO_2$. SQFZ (Pharmaceutical factory Batch No.180820) was obtained from Lizhu group Limin pharmaceutical Co. Ltd. (Guangdong, China). *Radix Astragali* and *Radix Codonopsis* are its main raw materials. The effective components are extracted and separated to obtain the intravenous preparation of traditional Chinese medicine, containing 160 g/L of the crude drug. Monoclonal antibodies against CD206, CD86, PI3K, P-gp, ABCG2, Bcl-2, Bax, and GAPDH were purchased from Cell Signaling Technology (Beverly, MA, USA).

### Co-culture protocol

MDA-MB-231/DDP and THP-1 cells were co-cultured using a cell culture insert (Corning, NY, USA) with a 0.4 μm porous membrane. The THP-1 cells ($5 \times 10^5$ cells/mL) were seeded in the upper chamber. PMA, 100 ng/mL (Sigma-Aldrich, USA), was used to stimulate cells for 24 h, and 10 ng/mL IL-4/IL-13 (Sino Biological, China) was used to stimulate cells for another 24 h. The upper chamber was washed thrice with phosphate-buffered saline (PBS). MDA-MB-231/DDP cells were seeded in the lower chamber at a density of $2.5 \times 10^5$ cells/mL for 24 h. Subsequently, THP-1-derived macrophages were placed on the top of the lower chamer containing MDA-MB-231/DDP cells. The co-cultured systems were incubated for 24 h in serum-free RPMI 1640 (Hyclone, USA).

### Tumour xenograft and treatment

Female nu/nu athymic mice (7 weeks of age) were obtained from Academia Sinica (Shanghai, China). MDA-MB-231/DDP cells ($2 \times 10^6$ cells) along with M2 macrophages ($4 \times 10^6$ cells) were injected into the right axillary flank of the mice. Cells were resuspended in 50% serum-free RPMI 1640 and 50% Matrigel medium in a total volume of 200 μl. Following the development of palpable tumours (approximately by 2 weeks), the mice were randomized into 9 groups (n = 8). The treatment groups included SQFZ (20, 40, and 60 mL/kg), cisplatin 2 mg/kg, SQFZ 40 mL/kg+ cisplatin 2 mg/kg, verapamil 1.35 mg/kg, and cisplatin 2 mg/kg+ verapamil 1.35 mg/kg. Untreated groups were divided into normal and controls, wherein each animal was injected with physiological saline. Tumour size was measured every five days, and the tumour volume was calculated as follows: V ($mm^3$) = L (major axis) × $W^2$ (minor axis)/2. After 30 days of treatment, blood was collected from the eyes, and the animals were sacrificed. Mice were euthanized by $CO_2$ inhalation. The maximum tumor volume permitted under animal protocols is 600 $mm^3$. Bottled compressed carbon dioxide gas with purity above 99% shall be used. Before the animals are placed in the euthanasia cage, the cage is not allowed to be filled with gas in advance. Instead, $CO_2$ should be filled at a balanced rate according to the $CO_2$ replacement rate of 30% - 70% of the container volume/minute, so as to make the animals

lose consciousness quickly and minimize their pain. During euthanasia, animals should lose consciousness within 2–3 minutes. The respiratory condition and eye colour of each animal should be continuously observed. After all animals stop breathing, $CO_2$ shall be injected for at least 1 minute. At the same time, the animals can only be taken out after observing that the animals have stopped breathing and their eyes have lost colour. After the euthanasia operation is completed, another appropriate method must be used to confirm the death of animals, such as cardiac arrest, respiratory arrest, animal stiffness and dilated pupils. Tumours were homogenized for western blot analysis. All procedures conformed to the guidelines of animal welfare and were approved by the ethical committee of Shanghai Traditional Chinese Medicine (approval number PZSHUTCM210402003).

## Western blot analysis

Cells were directly lysed in lysis buffer containing 2 mol/L sodium chloride, 10% NP-40, 10% SDS, 1 mol/L Tris-Cl, 1 g/L phenyl-methylsulfonyl fluoride, 0.1 g/L aprotinin, and 0.01 g/L leupeptin. The cell lysates were subjected to SDS-PAGE and then transferred onto polyvinylidene fluoride (PVDF) membranes. After the membranes were blocked with BSA for 1 h, the protein bands were detected using primary antibodies (1:1000) and secondary antibodies (1:800) conjugated with horseradish peroxidase and enhanced chemiluminescence reagents (Pharmacia, Buckinghamshire, UK). Western blots were quantitatively analysed using the Alpha Ease FC (FluorChem FC2) software. The band intensities of proteins of interest to GAPDH as the spot density were calculated using analytical tools.

## Enzyme linked immunesorbent (ELISA) assay

Serum was harvested and quantified by the Bradford method (Bio-Rad, Hercules, CA; $n = 8$ per group). Mouse serum PGE2, IL-6, IL-10, and CCL1 levels were quantified using an ELISA kit according to the manufacturer's directions (eBioscience, San Diego, CA).

## MTT assays

The co-cultured cells were plated onto 96-well plates in a cell culture medium containing various concentrations of drugs. Cell viability was assessed using the MTT assay as described previously (Promega, Madison, WI). Cytotoxicity was expressed as the percentage of surviving cells: $OD_{sample}/ OD_{control} \times 100\%$.

## Flow cytometric analysis

Cells ($10^6$/mL) were cultured in 6-well plates and reached 70–80% confluence after 6 h. The media containing FCS and insulin were not changed, and these cells were treated with SQFZ 100 μl/mL, LY294002 20 μmol/L, IGF-1 100 ng/mL, or the combination of SQFZ 100 μl/mL and IGF-1 100 ng/mL. Cells were subjected to an annexin V-PI dual staining assay following the manufacturer's protocol. Stained cells were analysed by fluorescence activating cell sorter (FACS) (Becton Dickinson, CA, USA), and the percentage of apoptotic cells was determined using the ModFit LT 3.0 software (Becton Dickinson, CA, USA).

## Statistical analysis

Data are expressed as the means±SD. The student's *t*-test was used to compare the differences between the two groups. One-way analysis of variance was used to analyze differences among three or more groups. Statistical significance was set at $P < 0.05$.

## Results

### SQFZ significantly improves the sensitivity of breast cancer cells toward cisplatin

Chemotherapeutic resistance often occurs in clinical treatment and improving the sensitivity of chemotherapeutic drugs is necessary. To determine the effects of combined SQFZ and cisplatin treatment on tumour growth, we established MDA-MB-231/DDP+M2 breast cancer xenografts. After 1 week, animals were injected i.p. daily with SQFZ 20, 40, or 60 mL/kg and cisplatin (2 mg/kg) for 4 weeks. After animals with MDA-MB-231/DDP+M2 cell xenografts developed sizable tumours, and were treated with SQFZ 20, 40, or 60 mL/kg and cisplatin (2 mg/kg) alone, respectively, the volumes of tumours were reduced by 17.95%, 33.37%, 37.01%, and 28.43%. In contrast, the combined treatment of SQFZ 40 mL/kg and cisplatin caused a 52.65% reduction in the tumour volume in 30 d. SQFZ (40 mL/kg) significantly enhanced the capability of cisplatin to reduce tumour mass (28.43–52.65%) (Fig 1A). Moreover, no apparent toxicity, as evidenced by animal body weights, was observed (Fig 1B). These results suggested that the combined treatment of SQFZ and cisplatin improved the sensitivity of MDA-MB-231/DDP+M2 cells in the primary xenograft.

### SQFZ affects the expression of TAMs

M2 TAMs affect the progression of tumours, which is closely related to drug resistance. To assess whether SQFZ could affect M2 TAMs, the expressions of CD206 and CD86 were detected in tumour grafts of mice receiving both SQFZ and cisplatin or each alone. The expression of CD206 decreased by 1.61- and 1.64-fold and CD86 increased by 1.54- and 1.55-fold following administration of SQFZ at 40 mL/kg and 60 mL/kg as compared to cisplatin (2 mg/kg) alone, respectively. The combined treatment of SQFZ 40 mL/kg and cisplatin 2 mg/kg led to CD 206 decrease by 1.89-fold and CD86 increase 1.76-fold relative to cisplatin 2 mg/kg alone, respectively (Fig 2). These results suggested that reduced cisplatin resistance following SQFZ administration was due to the decrease in M2-like macrophages and a concomitant increase in M1-like macrophages.

### SQFZ affects the levels of related cytokines

SQFZ can enhance the chemosensitivity of cisplatin by affecting the phenotype of TAMs. Therefore, we observed the effects of SQFZ on the levels of serum cytokines. The levels of PGE2, IL-6, and CCL1 decreased by 22.1%, 24.2%, and 27.3%, respectively, following the combination treatment of SQFZ and cisplatin relative to cisplatin alone. However, the level of IL-10 increased markedly by 20.1% after the combination treatment (Fig 3). These results suggested that SQFZ-mediated improved cisplatin resistance may be related to the release of inflammatory mediators by TAMs.

### SQFZ regulates the expression of chemoresistance-associated proteins

To investigate the potential mechanisms by which SQFZ improved the chemosensitivity of cisplatin, we first determined the levels of p-PI3K, PI3K, P-gp, and ABCG2 in MDA-MB-231/DDP+M2 xenografts following treatment. There was no significant difference in the expression of PI3K. The activation of p-PI3K decreased by 33.6% following the combination treatment with SQFZ and cisplatin as compared to cisplatin alone. Moreover, the levels of P-gp and ABCG2 decreased by 36.7% and 38.6% (Fig 4). These results suggested that SQFZ exerted a tumoricidal effect against cisplatin by decreasing the activation of the PI3K signalling pathway and inhibiting the expressions of P-gp and ABCG2.

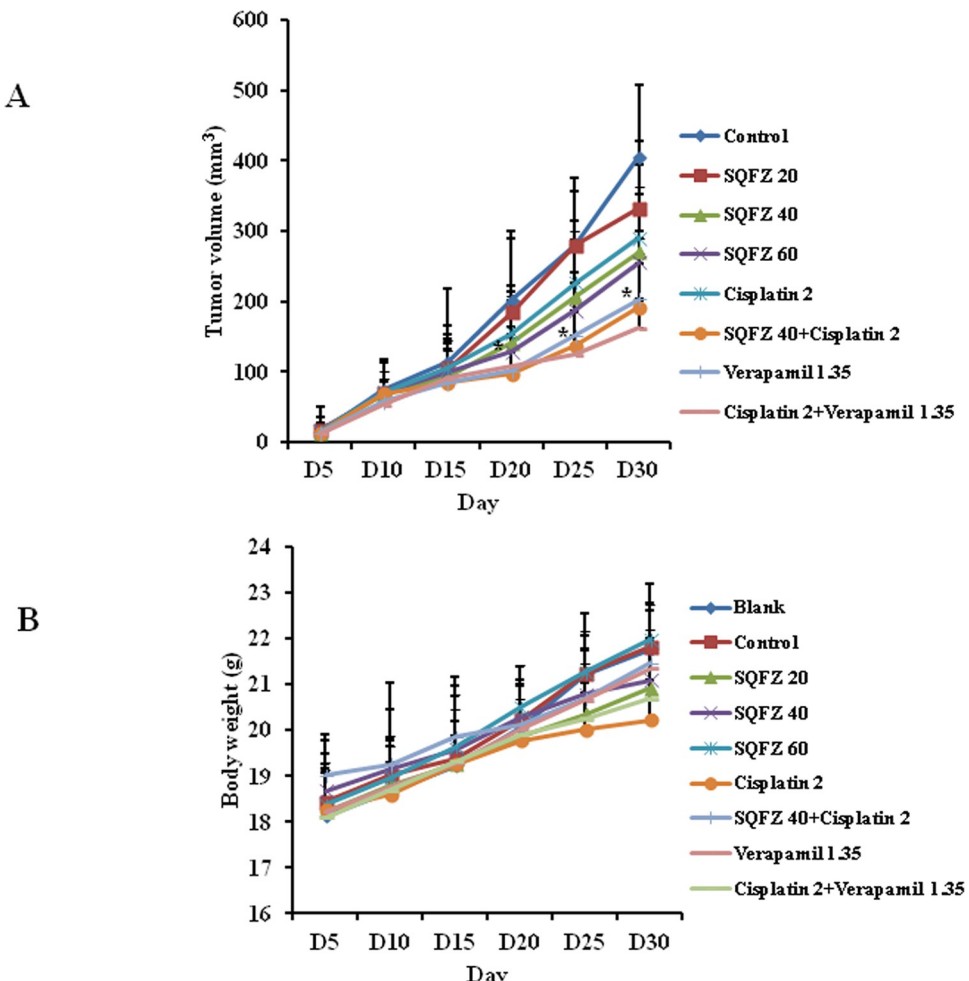

**Fig 1. Effects of SQFZ or SQFZ + cisplatin on tumour growth in MDA-MB-231/DDP+M2 xenografts.** BALB/c nude mice bearing MDA-MB-231/DDP+M2 in fat pads as xenografts were treated daily with i.p. injections of control or SQFZ 20, 40, 60 mL/kg, Cisplatin 2 mg/kg, SQFZ 40 mL/kg + Cisplatin, Verapamil 1.35 mg/kg, or Verapamil + Cisplatin for 4 weeks. Tumour volumes (A) and mouse body weights (B) were measured as described. Data are presented as the means ± SD (n = 10). *P < 0.05 compared with Cisplatin alone.

## SQFZ significantly improves the sensitivity in MDA-MB-231/DDP cells and M2 macrophages toward cisplatin

SQFZ improves the tumoricidal effects of cisplatin in vivo. To determine the effects of SQFZ on M2 macrophages, we assessed the cytotoxic effects of SQFZ on MDA-MB-231/DDP cells with or without M2 macrophages. Cells were exposed to varying concentrations of SQFZ for 48 h (Fig 5A). Cell survival was determined by the MTT assay. SQFZ resulted in an IC50 of 100 μl/mL in MDA-MB-231/DDP cells with M2 macrophages. The cell survival rate decreased by 39.6% as compared to the group with the same concentration in MDA-MB-231/DDP cells alone. The results suggested that SQFZ could inhibit MDA-MB-231/DDP cell growth related to M2 macrophages. When cisplatin was used in combination with SQFZ 40 μl/mL, the survival in the cisplatin 60 μM and 120 μM groups reduced by 50.8% and 72.5%, respectively, in the co-cultured MDA-MB-231/DDP cells and M2 macrophages (Fig 5B). These results suggested that SQFZ improved the sensitivity of cisplatin through M2 macrophages.

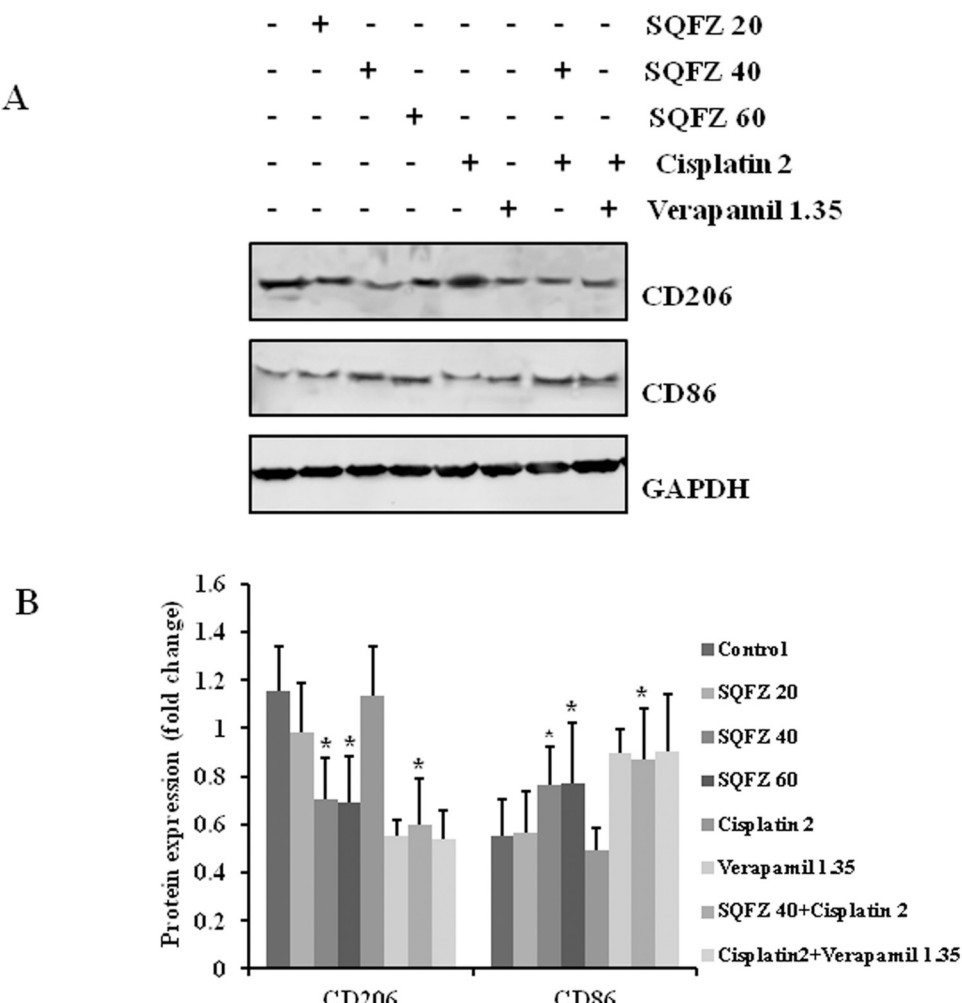

**Fig 2. Effects of SQFZ or SQFZ + Cisplatin on the expressions of CD206 and CD86 in MDA-MB-231/DDP+M2 xenografts.** Tumour lysates were analysed for the expression of CD206 and CD86 using the respective antibodies. The density ratio of detected proteins to GAPDH is shown as the relative expression. Values are shown as the means ± SD from three independent experiments. $^*p < 0.05$ as compared to Cisplatin alone.

## SQFZ affects cell growth through the PI3K signalling pathway in MDA-MB-231/DDP cells co-cultured with M2 macrophages

SQFZ inhibited the activation of PI3K in vivo and affected the growth of MDA-MB-231/DDP+M2 cells. To examine whether cell death was related to the PI3K signalling pathway induced by SQFZ, we performed MTT and flow cytometry assays on MDA-MB-231/DDP+M2 cells to quantitate the number of apoptotic cells. To ascertain whether the reversal of drug resistance by SQFZ in breast cancer cells correlated with activation of the PI3K signalling pathway, MDA-MB-231/DDP+M2 cells were treated with a PI3K-specific inhibitor (LY294002) or an activator (IGF-1). As expected, LY294002 or SQFZ inhibited cell proliferation (Fig 6A) and induced apoptosis (Fig 6B). However, SQFZ combined with IGF-1 weakened the reversal of resistance relative to the SQFZ- or LY294002-exposed groups alone (Fig 6A and 6B). These results suggested that SQFZ-induced apoptosis may be related to PI3K signalling.

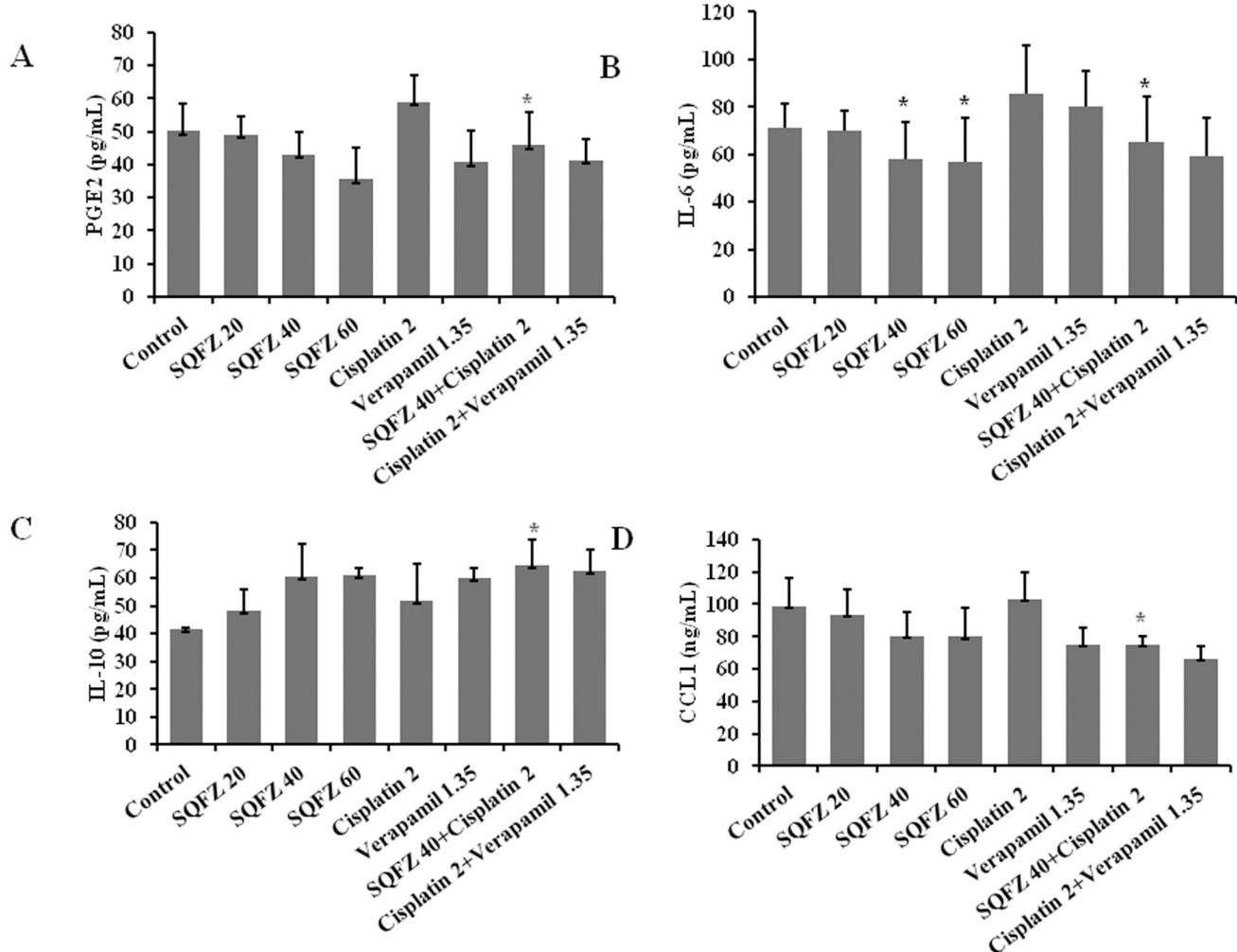

**Fig 3. Effects of SQFZ or SQFZ + Cisplatin on the levels of PGE2, IL-6, IL-10, and CCL1 in MDA-MB-231/DDP+M2 xenografts.** Serum was analysed for the levels of PGE2, IL-6, IL-10 and CCL1 by ELISA. Values are shown as the means ± SD from three independent experiments. *p < 0.05 as compared to Cisplatin alone.

## SQFZ inhibits resistance-associated and apoptosis-related proteins through the PI3K signalling pathway

To further determine the effects of SQFZ through PI3K, the levels of the ABC transporters, P-gp and ABCG2, and apoptosis-associated proteins, Bcl-2 and Bax, were detected by western blotting. The expressions of P-gp, ABCG2, and Bcl-2 decreased substantially, and that of Bax increased in both SQFZ and LY294002 alone treatment groups. Moreover, the levels of P-gp, ABCG2, and Bcl-2 were downregulated by the combination treatment of SQFZ and IGF-1 relative to IGF-1 alone. We also observed upregulation of Bax expression (Fig 7). These results suggested that the inhibition of P-gp, ABCG2, and Bcl-2 and the induction of Bax by SQFZ suppressed the activation of the PI3K signalling pathway.

## Discussion

The purpose of this study was to demonstrate whether SQFZ improved chemotherapeutic resistance induced by cisplatin through the PI3K signalling pathway in drug-resistant breast

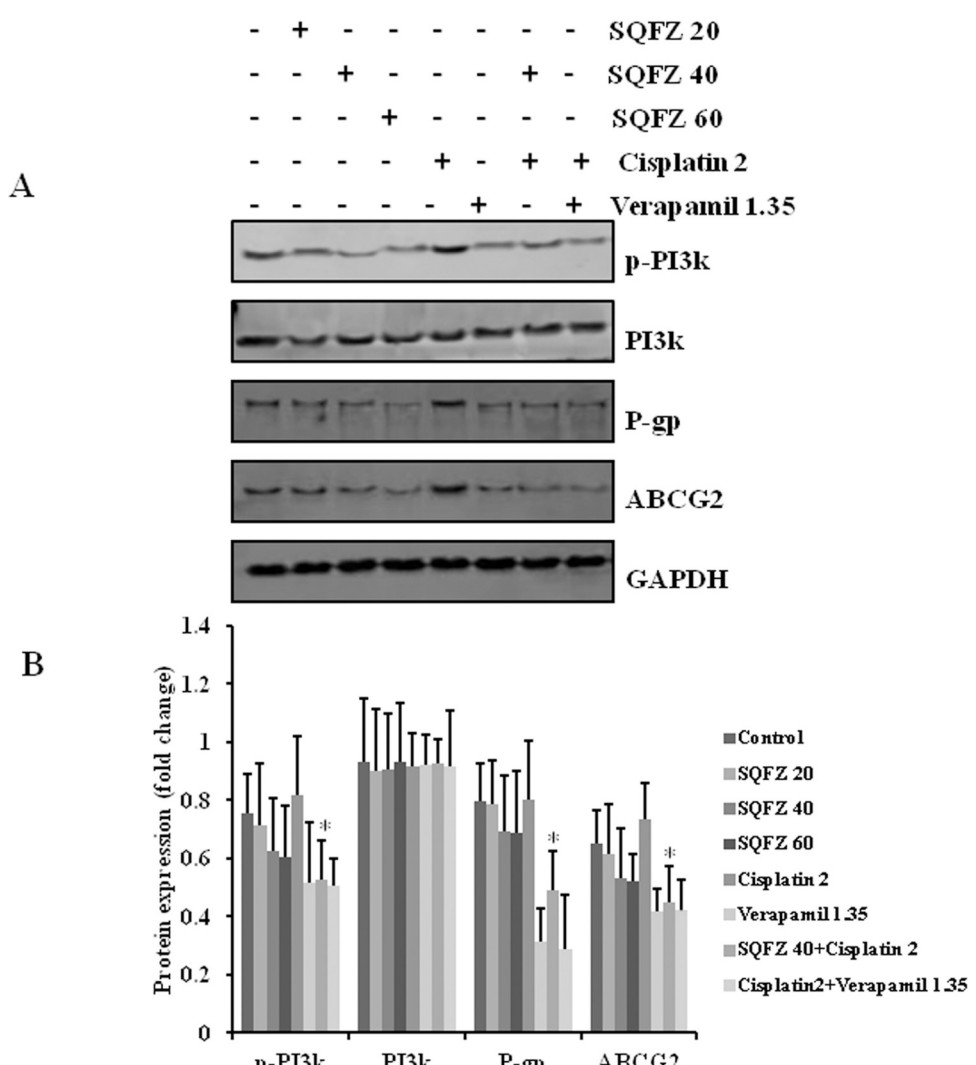

**Fig 4. Effects of SQFZ or SQFZ + Cisplatin on the expressions of p-PI3K, PI3K, P-gp, and ABCG2 in MDA-MB-231/DDP+M2 xenografts.** Tumour lysates were analysed for the expressions of p-PI3K, PI3K, P-gp, and ABCG2 with the respective antibodies. The density ratio of detected proteins to GAPDH is shown as relative expression. Values are shown as the means ± SD from three independent experiments. *p < 0.05 as compared to Cisplatin alone.

cancer. SQFZ improves clinical efficacy, inhibits immunosuppression, and reduces toxicity in combination with chemotherapy drugs for breast cancer [26, 27]. In the present study, we investigated the anticancer mechanisms of SQFZ action, particularly the signalling pathways involved in the reversal of drug resistance.

In the clinic, chemotherapy is the mainstay treatment for breast cancer, and chemotherapeutic resistance is the main reason for therapeutic failure [28]. Several factors can affect drug resistance, including the efflux of intracellular drugs and the regulation of apoptosis [29]. Increased expression of drug efflux pumps, including P-gp, and apoptosis have been linked to the development of multidrug resistance (MDR) [30]. Moreover, the therapeutic resistance is related to TAMs in some solid tumour types, including breast cancer [31]. TAMs promote oxaliplatin resistance in colorectal cancer [32]; M2-polarized TAMs are responsible for cancer resistance to cisplatin [33], and chemokines secreted by TAMs activate the PI3K/Akt/mTOR

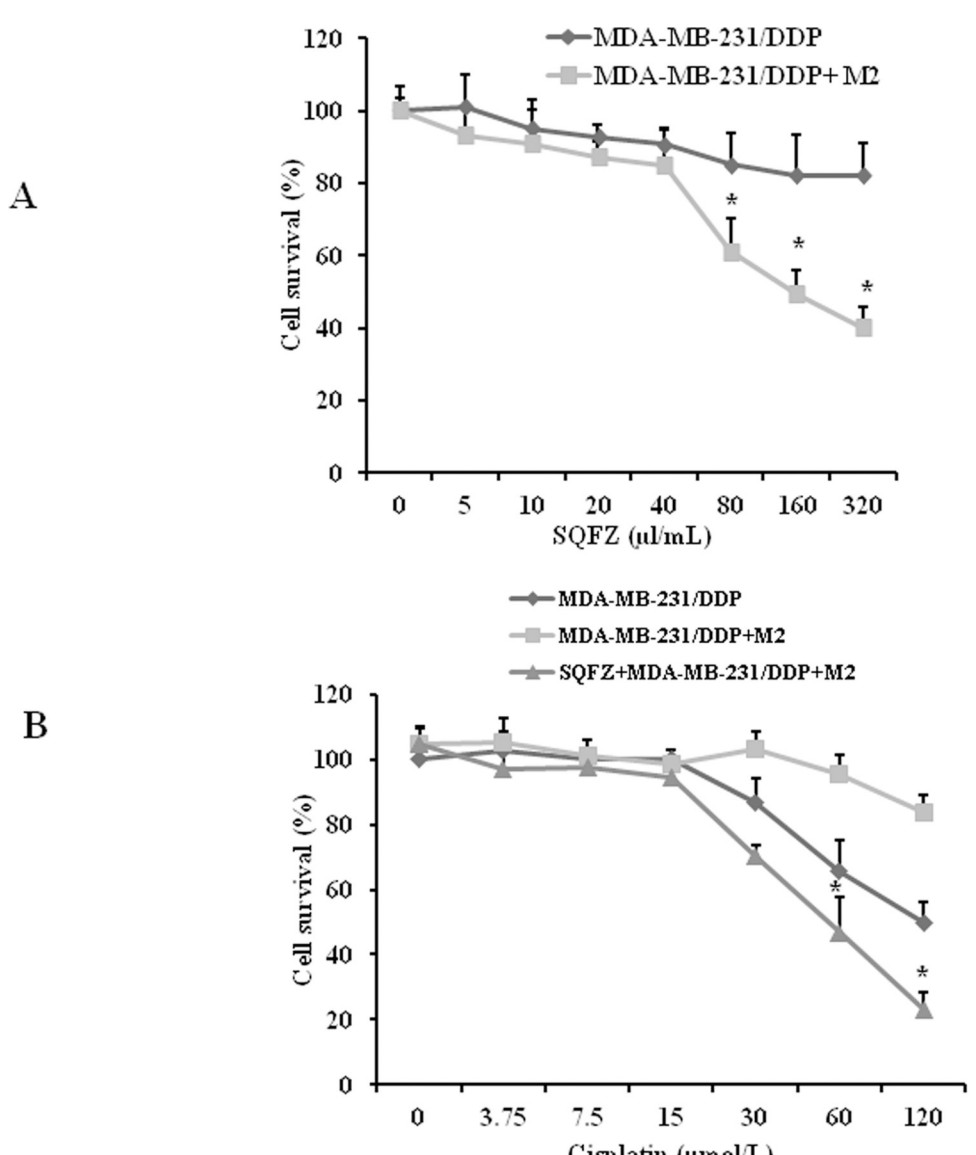

**Fig 5. Sensitivities of MDA-MB-231/DDP cells to SQFZ and SQFZ + Cisplatin.** (A) Both MDA-MB-231/DDP and MDA-MB-231/DDP co-cultured with M2-macrophage cells were incubated in SQFZ at 5, 10, 20, 40, 80, 160, or 320 μl/mL for 48 h. (B) MDA-MB-231/DDP and MDA-MB-231/DDP+M2 were treated with cisplatin at 3.75, 7.5, 15, 30, 60, or 120 μmol/L with or without SQFZ (40 μl/mL) for 48 h. Values are shown as the means ± SD from three independent experiments. *p < 0.05 as compared to Cisplatin alone.

signalling pathway, thereby promoting resistance [34]. In many MDR cancer cells, ABC membrane transporters are over expressed [35]. The PI3K signalling pathway is involved in MDR in various cancers. ABC membrane transporters reduce and drug resistance can be reversed following the inhibition of the PI3K signalling pathway [36].

Traditional Chinese medicine plays an important role in chemotherapy. It can improve the quality of life of patients, prolong survival, reduce side effects, and help treat cancer. SQFZ is an effective traditional Chinese medicine in clinical settings, and its treatment principle underlies healthy qi. Therefore, it is necessary to understand the molecular mechanism of chemoresistance by SQFZ.

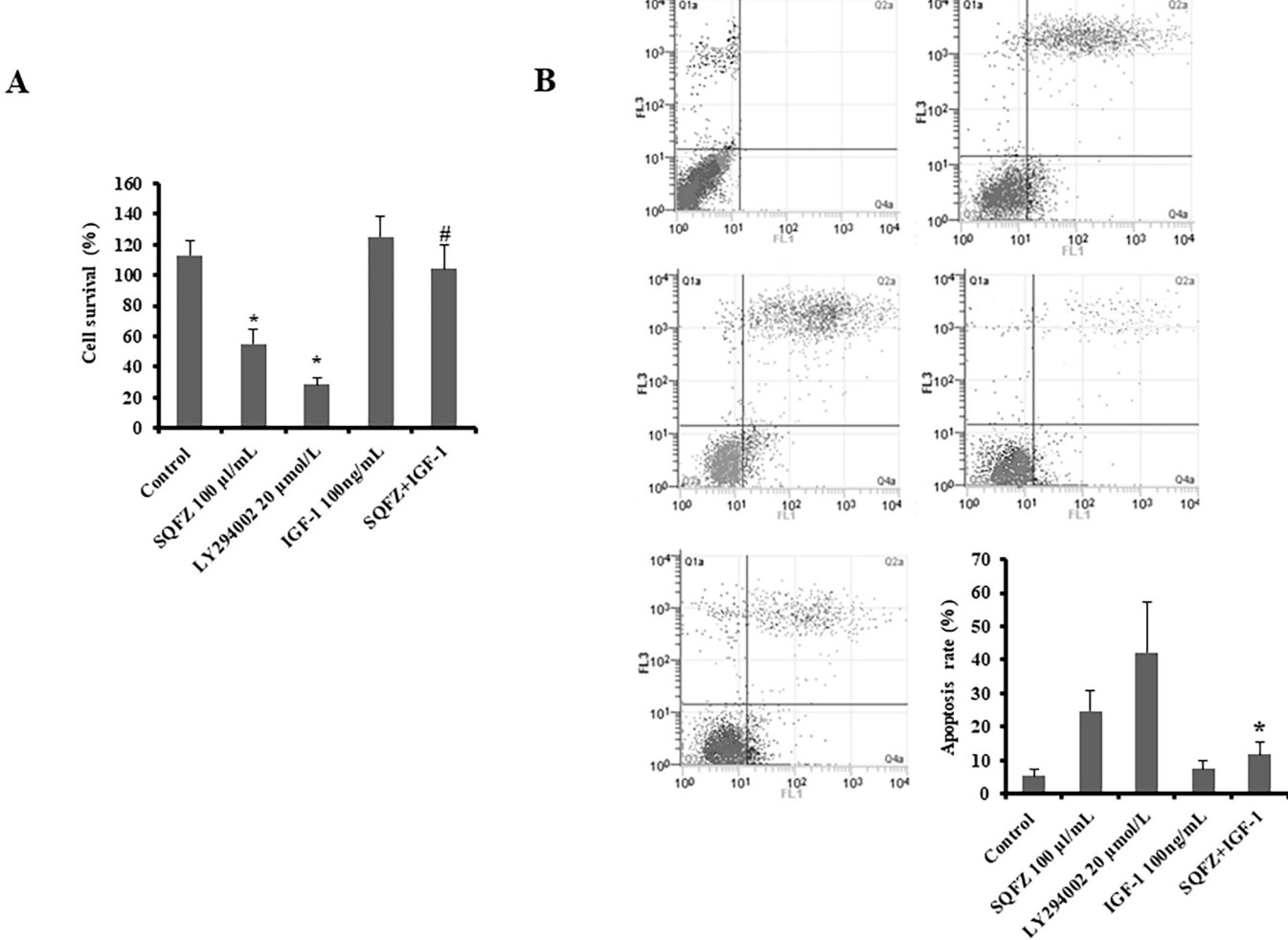

**Fig 6. SQFZ reverses constitutive and PI3K pathway-induced cisplatin resistance.** (A) Survival of MDA-MB-231/DDP+M2 cells. Cells were treated with SQFZ (100 μl/mL, 48 h), LY294002 (20 μmol/L, 2 h), IGF-1 (100 ng/mL, 48 h), and the combination of SQFZ (100 μl/mL, 48 h) and IGF-1 (100 ng/mL, 48 h) and subjected to the MTT assay. Values are shown as the means ± SE from three independent experiments. *p < 0.05 as compared to the Control. #p < 0.05 as compared to SQFZ 100 μl/mL alone. (B) Flow cytometry analysis of apoptosis based on Annexin V-FITC/PI binding to MDA-MB-231/DDP+M2 cells treated with SQFZ (100 μl/mL, 48 h), LY294002 (20 μmol/L, 2 h), IGF-1 (100 ng/mL, 48 h), and the combination of SQFZ (100 μl/mL, 48 h) and IGF-1 (100 ng/mL, 48 h).

SQFZ, as adjunctive therapy in cancer treatment, is effective in clinical settings. In the present study, SQFZ improved the sensitivity toward cisplatin in vitro and in vivo (Figs 1 & 5). However, the mechanism of SQFZ action in cisplatin resistance remains unknown. Earlier reports suggest that traditional Chinese medicine treatment with Fuzheng Jiedu Formula (FZJD), Gancao (Glycyrrhiza Radix et Rhizoma), Renshen (Ginseng Radix et Rhizoma), or baicalein reduces M2-TAMs [37]. In this study, a remarkable increase in the levels of CD206 and a decrease in CD86 were observed in xenografts (Fig 2). These results indicated that SQFZ could regulate the ratio of M2 to M1 cells. SQFZ decreased the levels of PGE2, IL-6, and CCL1 and increased that of IL-10 induced by cisplatin (Fig 3). The effect of SQFZ may be related to its regulation of the levels of inflammatory factors and chemokines. To investigate whether the effects of SQFZ on reducing cisplatin resistance were related to M2 cells, we assessed the sensitivity of cisplatin treatment by SQFZ in MDA-MB-231/DDP and M2 co-cultured cells. The survival of co-cultured cells was significantly lower than that of cells treated with MDA-MB-

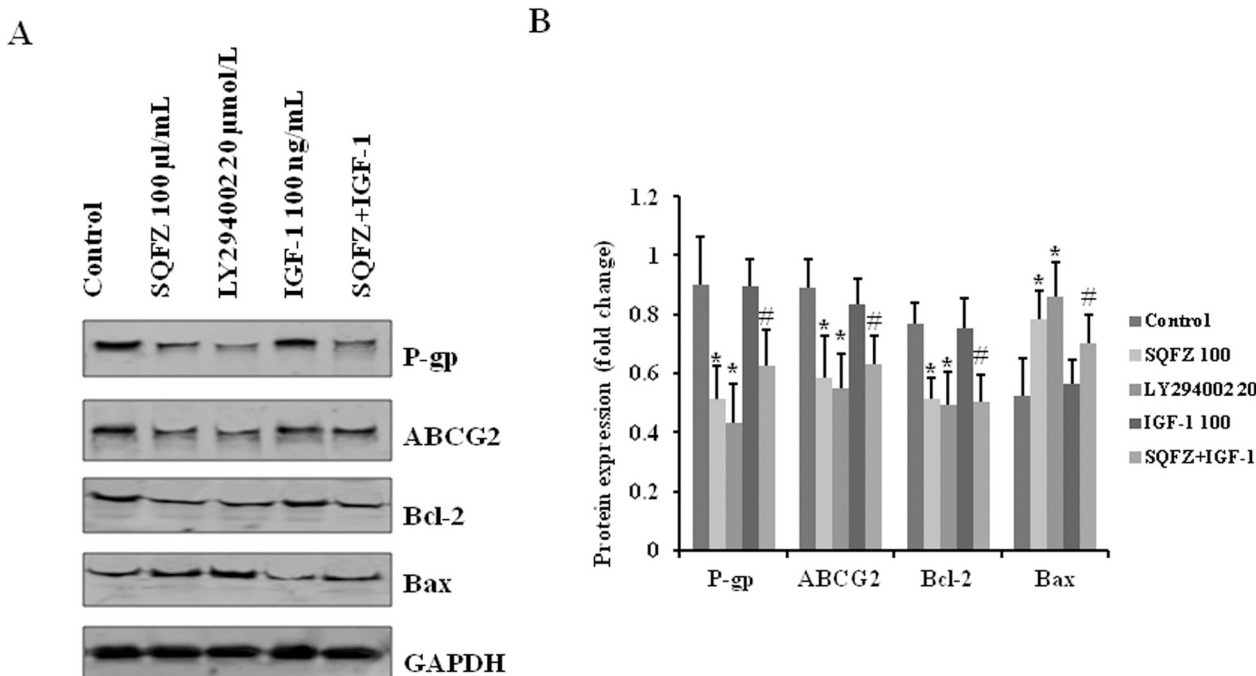

**Fig 7. SQFZ inhibits the expressions of P-gp, ABCG2, and Bcl-2 and increases that of Bax through the PI3K pathway.** Cell lysates were analysed for the expressions of P-gp, ABCG2, Bcl-2, and Bax with the respective antibodies. The density ratio of detected proteins to GAPDH is shown as relative expression. Values are shown as the means ± SD from three independent experiments. *p < 0.05 as compared to the Control. #p < 0.05 as compared to IGF-1 alone.

231/DDP alone. Moreover, SQFZ reduced the IC50 concentration of cisplatin in MDA-MB-231/DDP and M2 co-cultured cells more significantly than in MDA-MB-231/DDP alone (Fig 5). These results suggested that M2 mediated the effects of SQFZ on enhancing chemosensitivity. To elucidate the mechanism of SQFZ-mediated alleviation of drug resistance, we assessed the expressions of drug resistance-related proteins. SQFZ inhibited the activity of PI3K and decreased the expressions of P-gp and ABCG2 in vivo (Fig 4). Subsequently, we blocked and promoted the expression of PI3K to observe whether SQFZ alleviated drug resistance through the PI3K signalling pathway in vitro. SQFZ indeed induced apoptosis through the PI3K signalling pathway and regulated the expression of drug resistance- and apoptosis-associated proteins (Figs 6 & 7). SQFZ is a compound preparation in traditional Chinese medicine, and its mechanisms may include multiple targets. We only studied one of the signalling pathways, and several others necessitate further investigations. Moreover, we only assessed the mechanism of SQFZ-mediated improvement in drug resistance through the regulation of M2-TAMs. The regulatory effects of SQFZ on other immune cells need to be further examined. We will explore the mechanisms of SQFZ's actions from multiple perspectives in the future.

## Conclusions

SQFZ is an effective prescription for cancer treatment in clinical settings [38, 39]. In summary, the present study showed that SQFZ could reverse M2-mediated cisplatin resistance through the PI3K pathway. This suggested that SQFZ may be a natural and potent inhibitor of drug resistance in breast cancer. Traditional Chinese medicine may yield better results in cancer treatment, and SQFZ necessitated further investigation as a potential therapeutic strategy for drug-resistant cancer.

## Supporting information

**S1 Raw images.**
(PDF)

## Acknowledgments

We are grateful to the technical team of Shanghai Traditional Chinese Medicine.

## Author Contributions

**Data curation:** Mei-na Ye.

**Formal analysis:** Dong-dong Fang.

**Investigation:** Mei-na Ye.

**Methodology:** Bin Yan, Rong Shi.

**Software:** Yi-yu Lu.

**Writing – original draft:** Qian-mei Zhou.

**Writing – review & editing:** Qian-mei Zhou.

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
