## [Decision Letter · Decision Letter 0]

16 Aug 2022

PONE-D-22-20060Shenqi Fuzheng Injection Reverses M2 Macrophages-Mediated Cisplatin Resistance through the PI3K Pathway in Breast Cancer TreatmentPLOS ONE

Dear Dr. zhou,

Thank you for submitting your manuscript to PLOS ONE. After careful consideration, we feel that it has merit but does not fully meet PLOS ONE’s publication criteria as it currently stands. Therefore, we invite you to submit a revised version of the manuscript that addresses the points raised during the review process. Please carefully address the reviewer's concerns, and the manuscript needs a deep editing by an English editor before resubmission. Please submit your revised manuscript by Sep 30 2022 11:59PM. If you will need more time than this to complete your revisions, please reply to this message or contact the journal office at plosone@plos.org. Please include the following items when submitting your revised manuscript:A rebuttal letter that responds to each point raised by the academic editor and reviewer(s). You should upload this letter as a separate file labeled 'Response to Reviewers'.A marked-up copy of your manuscript that highlights changes made to the original version. You should upload this as a separate file labeled 'Revised Manuscript with Track Changes'.An unmarked version of your revised paper without tracked changes. You should upload this as a separate file labeled 'Manuscript'.

We look forward to receiving your revised manuscript.

Kind regards,

Ming Tan

Academic Editor

PLOS ONE

Journal Requirements:

3. To comply with PLOS ONE submissions requirements, in your Methods section, please provide additional information regarding the experiments involving animals and ensure you have included details on (1) methods of sacrifice, (2) methods of anesthesia and/or analgesia, and (3) efforts to alleviate suffering.

Reviewers' comments:

Reviewer's Responses to Questions

**Comments to the Author**

1. Is the manuscript technically sound, and do the data support the conclusions?

Reviewer #1: Partly

2. Has the statistical analysis been performed appropriately and rigorously? 

Reviewer #1: Yes

3. Have the authors made all data underlying the findings in their manuscript fully available?

Reviewer #1: No

4. Is the manuscript presented in an intelligible fashion and written in standard English?

Reviewer #1: No

5. Review Comments to the Author

Reviewer #1: The authors showed the Shenqi Fuzheng injection could reverse M2 macrophages-mediated cisplatin resistance through the PI3K pathway in breast cancer treatment. There are few major flaws in the manuscript which might lead to difficulty in understanding the whole study.

1. The preparation of SQFZ is missing. How and where to obtain the SQFZ not stated. Identification of the SQFZ also not provided. These are all important information as it provides details information for others in case they interested to study further on this SQFZ.

2. Animal husbandry information missing and kindly provide animal ethic reference number in the methodology section.

3. What is DDP? The function of DDP was not mentioned in the manuscript.

4. Due to the cell MDA-MB-231 was donated by the hospital, and it is important to do the characterization and identification of the cell via flow cytometry. This was missing in the manuscript.

5. The methodology especially "Coculture procedure" somehow is very confusing and unclear. Is both cells coculture at the same time or THP-1 cell to be cultured first then only co-culture with MDA-MB-231? Is both cells using the same media with DDP?

6. Why the cells need to be stimulated twice and with difference inducers, i.e., PMA and IL-4/IL-13?

7. What is M2 macrophages cell? How to culture it? There is no information that related to M2 macrophages cell at all in the methodology.

8. Regarding the MTT assay, what type of cell going to be tested here? No information about the cells in this section.

9. According to the section results, " SQFZ significantly improves the sensitivity to cisplatin in MDA-MB-231/DDP cells and M2 macrophages", the authors stated that cytotoxicity effect detected on MDA-MB-231/DDP cells with or without M2 macrophages upon SQFZ treatment. It is kind of confusing here as the author never mention that both cells will be coculture and test for cytotoxicity effect. The main problem is how to differentiate the cell death between MDA and macrophages cell within the same well? The methodology itself seems rise a major concern here.

10. The manuscript need to send for English editing service as there were many grammatically errors and structure problem.

6. PLOS authors have the option to publish the peer review history of their article (what does this mean?). If published, this will include your full peer review and any attached files.

Reviewer #1: No

---

## [Author Response · Author response to Decision Letter 0]

23 Nov 2022

Dear editor and reviewers,

On behalf of my co-authors, we thank you very much for giving us an opportunity to revise our manuscript, we appreciate you and reviewers very much for their positive and constructive comments and suggestions on our manuscript entitled “Shenqi Fuzheng Injection Reverses M2 Macrophage-Mediated Cisplatin Resistance through the PI3K Pathway in Breast Cancer” (PONE-D-22-20060). Those comments are all valuable and very helpful for revising and improving our paper, as well as the important guiding significance to our researches. We have studied comments carefully and have made correction. The main corrections in the paper and the responds to the reviewer’s comments are as flowing:

Responds to the reviewer’s comments:

1. The preparation of SQFZ is missing. How and where to obtain the SQFZ not stated. Identification of the SQFZ also not provided. These are all important information as it provides details information for others in case they interested to study further on this SQFZ.

Response: SQFZ (Pharmaceutical factory Batch No.180820) was obtained from Lizhu group Limin pharmaceutical Co. Ltd.( Guangdong, China). Radix Astragali and Radix Codonopsis are its main raw materials. The effective components are extracted and separated to obtain the intravenous preparation of traditional Chinese medicine, containing 160 g/L of the crude drug. These have been added in the manuscript and marked in red.

2. Animal husbandry information missing and kindly provide animal ethic reference number in the methodology section.

Response: All procedures conformed to the requirements of animal welfare and were approved by the ethical committee of Shanghai Traditional Chinese Medicine (approval number PZSHUTCM210402003). The animal ethic reference number has been added.

3. What is DDP? The function of DDP was not mentioned in the manuscript.

Response: DDP is cisplatin. Cisplatin (DDP) is a major chemotherapeutic agent used for the treatment of triple-negative breast cancer (TNBC). However, a risk of developing resistance to cisplatin treatment exists, and leads to treatment failure. It has been added in the manuscript.

4. Due to the cell MDA-MB-231 was donated by the hospital, and it is important to do the characterization and identification of the cell via flow cytometry. This was missing in the manuscript.

Response: The hospital has identified the cells before donating them to us. The intracellular concentration of DDP was determined by HPLC in MDA-MB-231 and MDA-MB-231 / DDP. Cells were incubated with DDP for 24 h. The concentration of DDP in MDA-MB-231 cells was (47.10 ± 2.37) ng / (5 × 104) cells. However, the concentration of DDP in MDA-MB-231 / DDP cells was (6.30 ± 1.64) ng / (5 × 104) cells. There was significant difference between them. 

5. The methodology especially "Coculture procedure" somehow is very confusing and unclear. Is both cells coculture at the same time or THP-1 cell to be cultured first then only co-culture with MDA-MB-231? Is both cells using the same media with DDP?

Response: The THP-1 cells were seeded in the upper chamber. PMA was used to stimulate cells for 24 h, and IL-4/IL-13 was used to stimulate cells for another 24 h. The upper chamber was washed thrice with phosphate-buffered saline (PBS). MDA-MB-231/DDP cells were seeded in the lower chamber for 24 h. Subsequently, the upper chamber containing THP-1-derived macrophages were placed on the top of the lower chamer containing MDA-MB-231/DDP cells. Only the lower chamer containing MDA-MB-231/DDP cells used the media with DDP. 

6. Why the cells need to be stimulated twice and with difference inducers, i.e., PMA and IL-4/IL-13?

Response: THP-1 is a human monocytic leukemia cell. THP-1-derived macrophages were characterized after stimulation with phorbol-12-myristate 13-acetate (PMA). IL-4/IL-13 induced macrophages polarizing toward M2 type. Therefore, the cells need to be stimulated twice and with difference inducers.

7. What is M2 macrophages cell? How to culture it? There is no information that related to M2 macrophages cell at all in the methodology.

Response: Macrophages are classified into two major phenotypes based on their functions—the M1 phenotype with proinflammatory responses and antitumor functions, whereas the M2 phenotype is anti-inflammatory and tumor-promoting. 

The cells were cultured in RPMI 1640 with 10% fetal bovine serum (FBS) and penicillin/streptomycin at 37℃ in a humidified 5% CO2 atmosphere. PMA was used to stimulate cells for 24 h, and IL-4/IL-13 was used to stimulate cells for another 24 h. 

8. Regarding the MTT assay, what type of cell going to be tested here? No information about the cells in this section.

Response: The cocultured cells of MDA-MB-231/DDP and M2 or MDA-MB-231/DDP cells alone were tested here. It has been added in the manuscript. To determine the effects of SQFZ on M2 macrophages, we assessed the cytotoxic effects of SQFZ on MDA-MB-231/DDP cells with or without M2 macrophages. Cell survival was determined by the MTT assay.

9. According to the section results, " SQFZ significantly improves the sensitivity to cisplatin in MDA-MB-231/DDP cells and M2 macrophages", the authors stated that cytotoxicity effect detected on MDA-MB-231/DDP cells with or without M2 macrophages upon SQFZ treatment. It is kind of confusing here as the author never mention that both cells will be coculture and test for cytotoxicity effect. The main problem is how to differentiate the cell death between MDA and macrophages cell within the same well? The methodology itself seems rise a major concern here.

Response: MDA-MB-231/DDP cells with or without M2 will be cocultured and tested for cytotoxicity effect. It has been added in the manuscript. The coculture systems we used were non-contact coculture. THP-1 cells in the upper chamber affected MDA-MB-231/DDP cells in the lower chamber by some secretions such as cytokines and chemokines. MDA-MB-231/DDP cells treated with SQFZ were tested for cytotoxicity effect. 

10. The manuscript need to send for English editing service as there were many grammatically errors and structure problem.

Response: We have worked on both language and readability and have also involved native English speakers for language corrections. We really hope that language level has been improved.

---

## [Editor Report · Decision Letter 1]

14 Dec 2022

Shenqi Fuzheng Injection Reverses M2 Macrophage-Mediated Cisplatin Resistance through the PI3K Pathway in Breast Cancer

PONE-D-22-20060R1

Dear Dr. zhou,

We’re pleased to inform you that your manuscript has been judged scientifically suitable for publication and will be formally accepted for publication once it meets all outstanding technical requirements.

Kind regards,

Ming Tan

Academic Editor

PLOS ONE

Additional Editor Comments (optional):

The authors have addressed all the concerns of the original reviewer and the manuscript has been improved.
---

## [Editor Report · Acceptance letter]

4 Jan 2023

PONE-D-22-20060R1 

Shenqi Fuzheng Injection Reverses M2 Macrophage-Mediated Cisplatin Resistance through the PI3K Pathway in Breast Cancer 

Dear Dr. Zhou:

I'm pleased to inform you that your manuscript has been deemed suitable for publication in PLOS ONE. Congratulations! Your manuscript is now with our production department. 

Kind regards, 

on behalf of

Dr. Ming Tan 

Academic Editor

PLOS ONE